# Increase in concerns about climate change following climate strikes and civil disobedience in Germany

Johannes Brehm [1,2] ✉ & Henri Gruhl [1,3] ✉

Climate movements have gained momentum in recent years, aiming to create public awareness of the consequences of climate change through salient climate protests. This paper investigates whether concerns about climate change increase following demonstrative protests and confrontational acts of civil disobedience. Leveraging individual-level survey panel data from Germany, we exploit exogenous variations in the timing of climate protests relative to survey interview dates to compare climate change concerns in the days before and after a protest (N = 24,535). Following climate protests, we find increases in concerns about climate change by, on average, 1.2 percentage points. Further, we find no statistically significant evidence that concerns of any subpopulation decreased after climate protests. Lastly, the increase in concerns following protests is highest when concern levels before the protests are low.

Raising awareness of the potentially devastating consequences of climate change is fundamental to addressing the climate crisis. Increasing public concerns about these consequences can lead to climate change being perceived as a pressing political issue. Elevated concerns can lead to changes in voting behavior, favoring political parties seeking to address climate change[1-4]. Moreover, increased awareness can translate into public support for ambitious mitigation policies[5-8].

Thus, a proclaimed goal of various climate movements is to raise public awareness of the consequences of climate change[9,10]. To achieve this goal, they organize global climate strikes and acts of civil disobedience, such as obstructing coal mines, power plants, and roads[11]. Here, we focus on three climate movements that employ different protest tactics. Fridays for Future (FFF) is a youth-led and -organized movement that organizes global climate strikes[12]. On the other hand, Ende Gelände (EG), a German anti-coal movement[13], and Extinction Rebellion (XR), an international and politically non-partisan movement[14], resort to direct action and non-violent civil disobedience tactics[14,15]. These are deliberate, public, and non-violent acts of protest that defy established laws[16,17]. To investigate differences in tactics, we classify global climate strikes as demonstrative protests and acts of civil disobedience as confrontational protests. Media coverage of climate protests serves as a mediator in shaping the protest's impact on a broader population, extending beyond those directly involved or affected by the protests[18,19]. However, it is a subject of intense public debate whether climate protests are effective in raising climate change concerns in the population or whether they may backfire, for instance, with confrontational acts of civil disobedience turning people away from the cause[20-23]. Thus far, only limited empirical evidence exists on how protests affect the share of people concerned about climate change[24].

Here, we investigate whether concerns about the consequences of climate change among the general public in Germany increase after climate protests. We base our empirical strategy on the quasi-random timing of survey interviews relative to the timing of climate protests, which enables us to estimate the causal effects of climate protests on climate change concerns in a before-after research design[25]. To identify salient climate protests, we collect data on news coverage of leading public service broadcasters in Germany (Table 1). We verify the salience of the protests using Google Trends (Fig. 1) and the number of articles on the movements in the six highest-circulated German daily newspapers (Supplementary Fig. 1 in the Supplementary Information file), which both spike during the week of the protests. Using this data, we exploit differences in the timing of survey responses in the German Socio-Economic Panel (SOEP), which has the great advantage that interviews are conducted throughout the entire year. Comparing individuals surveyed in the days before a climate protest with those

[1]RWI - Leibniz Institute for Economic Research, Berlin Office, Berlin, Germany. [2]Hertie School, Berlin, Germany. [3]Vrije Universiteit Amsterdam, Amsterdam, Netherlands. ✉e-mail: johannes.brehm@rwi-essen.de; henri.gruhl@rwi-essen.de

**Table 1 | Salient climate protests in Germany, 2016–2020**

| Confrontational protests | | | | | | | | Demonstrative protests | | | |
|---|---|---|---|---|---|---|---|---|---|---|---|
| **Ende Gelände (EG)** | | | | **Extinction Rebellion (XR)** | | | | **Fridays for Future (FFF)** | | | |
| 13 | May | 2016 | Blocking coal mine, Lusatia[1] | 07 | Oct | 2019 | Road blocking & climate camp, Chancellery Berlin[10] | 15 | Mar | 2019 | Global Climate Strike[6] |
| 26 | Aug | 2017 | Blocking coal mine, Rhenish mining area[2] | 01 | Jul | 2020 | Demonstration, Party headquarters Berlin[14] | 24 | May | 2019 | Global Climate Strike[7] |
| 04 | Nov | 2017 | Demonstration & blocking coal mine, Hambach[3] | | | | | 20 | Sep | 2019 | Global Climate Strike[9] |
| 06 | Oct | 2018 | Demonstration Hambach forest[4] | | | | | 29 | Nov | 2019 | Global Climate Strike[11] |
| 27 | Oct | 2018 | Blocking coal mine, Hambach[5] | | | | | 24 | Apr | 2020 | Global Climate Strike[13] |
| 22 | Jun | 2019 | Blocking coal mine, Rhenish mining area[8] | | | | | 25 | Sep | 2020 | Global Climate Strike[16] |
| 30 | Nov | 2019 | Blocking coal mine, Lusatia[11] | | | | | | | | |
| 02 | Feb | 2020 | Blocking power plant, Datteln[12] | | | | | | | | |
| 30 | Aug | 2020 | Demonstration, Rhenish mining area[15] | | | | | | | | |
| 23 | Sep | 2020 | Climate camp, Hoher Busch[16] | | | | | | | | |
| 22 | Nov | 2020 | Blocking deforestation, Dannenröder forest[17] | | | | | | | | |

Climate protests in Germany that appeared in the evening news of the two leading public service broadcasters (ARD and ZDF, see "Method" section).

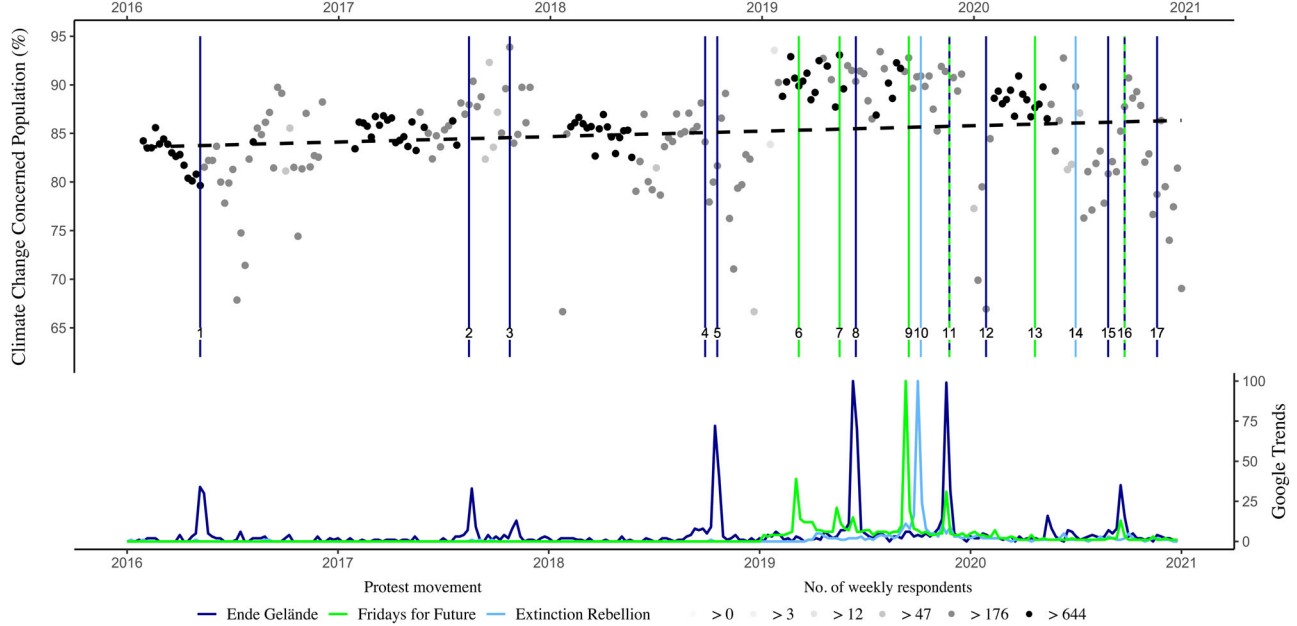

**Fig. 1 | Climate change concerns in Germany and Google Trends of protest movements.** Dots show the weekly averages of concerns about the consequences of climate change constructed as a binary variable (not concerned or concerned) from 2016 to 2020 (SOEP) between 65 and 95 percent. Darker shades of gray indicate a higher number of observations per week. The dashed line represents a linear fit, while the vertical lines indicate the dates of the 17 identified climate protests (see numbers corresponding to Table 1). The graph below shows the weekly values of Google Trends for Fridays for Future (FFF), Ende Gelände (EG), and Extinction Rebellion (XR). Source data are provided as a Source Data file.

surveyed in the days thereafter allows us to estimate causal effects. Our main and most restrictive sample consists of 24,535 observations and spans from 2016 to 2020, the latest available SOEP survey wave.

Concerns about climate change have increased considerably in recent years (Fig. 1), with climate protests being discussed as one of the drivers[18,24,26]. Given the large number of observations, we can focus on small time windows around protests, allowing us to identify their effect without picking up the general long-term trend of rising climate change concerns. Thus, we can address concerns about reverse causality in this research field since it is also plausible that increased climate change concerns lead to more climate protests[4,25]. In addition, our method allows us to assess whether climate protests merely preach to the converted or reach those previously not concerned about climate change[18]. Lastly, including various climate movements with different tactics (i.e., peaceful marches and acts of civil disobedience), we

contribute to the current debate about successful and effective strategies for climate movements and whether some protest forms may decrease concerns about climate change.

## Results

### Concerns about climate change increase following climate protests

Our results show that the probability that a respondent is concerned about the consequences of climate change increase on average by 1.2 percentage points in the 14 days after a protest (Table 2, Column (6), LPM). The effect is statistically significant at the five percent level ($p = 0.012$, degrees of freedom (df) = 24,185). In other words, a higher share of the population is concerned about climate change after a protest. This result indicates that climate protests do not merely preach to the converted but also convince previously unconcerned

**Table 2 | Main effect of climate protests on climate change concerns**

| | | Dependent variable: climate concern (0–1 dummy) | | | | | |
|---|---|---|---|---|---|---|---|
| Coefficient | Model | (1) | (2) | (3) | (4) | (5) | (6) |
| Post protest | Linear Probability (LPM) | 0.0122 (0.0065) [0.0786] | 0.0127 (0.0047) [0.0150] | 0.0127 (0.0048) [0.0166] | 0.0130 (0.0048) [0.0146] | 0.0128 (0.0046) [0.0136] | 0.0119 (0.0042) [0.0120] |
| Post protest | LPM with entropy balancing | 0.0117 (0.0055) [0.0506] | 0.0138 (0.0046) [0.0086] | 0.0140 (0.0047) [0.0095] | 0.0142 (0.0048) [0.0088] | 0.0139 (0.0046) [0.0078] | 0.0138 (0.0043) [0.0053] |
| Post protest | Probit | 0.0117 (0.0058) [0.0422] | 0.0133 (0.0043) [0.0019] | 0.0134 (0.0044) [0.0021] | 0.0138 (0.0044) [0.0019] | 0.0136 (0.0044) [0.0021] | 0.0128 (0.0043) [0.0025] |
| Protest fixed effects | | ✓ | ✓ | ✓ | ✓ | ✓ | ✓ |
| Year fixed effects | | ✓ | ✓ | ✓ | ✓ | ✓ | ✓ |
| Individual controls | | ✓ | ✓ | ✓ | ✓ | ✓ | ✓ |
| Calendar month fixed effects | | | ✓ | ✓ | ✓ | ✓ | ✓ |
| Weekday fixed effects | | | ✓ | ✓ | ✓ | ✓ | ✓ |
| Weather controls | | | | ✓ | ✓ | ✓ | ✓ |
| Elections/COPs controls | | | | ✓ | ✓ | ✓ | ✓ |
| Interviewer controls | | | | | ✓ | ✓ | ✓ |
| State fixed effects | | | | | | ✓ | ✓ |
| Protest×State fixed effects | | | | | | | ✓ |
| Number of observations | | 24,566 | 24,566 | 24,541 | 24,541 | 24,541 | 24,535 |

Main results using linear probability and probit models across specifications (see "Methods" section). With Entropy balancing weights, the sample size decreases to $N = 22,994$ (1-5), and 22,987 (6). Marginal effects are reported for the probit estimation with $N = 24,565$ (1-2), 24,540 (3-5), and 24,266 (6). Robust standard errors are clustered at the protest level and in parentheses. $P$ values are based on two-tailed t-tests and in square brackets.

individuals. An average effect of 1.2 percentage points is notable given the already high share of people in Germany concerned about climate change (81 percent in 2015). To put this finding into perspective, Hoffmann et al.[1] find that environmental concerns increase by 0.5–0.8 percentage points in response to a one standard deviation increase in a dry spell or temperature anomaly, respectively. Our relatively large effect of climate protests aligns with the findings of Sisco et al.[18], who find that global climate marches are more powerful drivers of attention to climate change than political events (UN COPs) and extreme temperatures.

We perform several tests to test the robustness of these findings. First, the main finding is corroborated by modeling the outcome as multiple concern levels instead of the binary variable (Supplementary Table 8). The likelihood that respondents are not concerned decreases (coef. = −1.24pp, $p = 0.003$), while we find no statistically significant effects at the five percent level on the likelihood of some concern (coef. = 0.58pp, $p = 0.489$) and high concern (coef. = 0.66pp, $p = 0.570$), but positive coefficients. This indicates that our results are driven by the extensive margin, i.e., whether or not a person is generally concerned with climate change and not how concerned they are (intensive margin). Second, we show that our main results in Table 2 are independent of the chosen estimation method. In the second row, we apply entropy balancing[27], a re-weighting scheme that improves the balance of covariate means between the treatment and control groups, and find that results stay qualitatively the same, with slightly larger coefficients. In the most restrictive specification, the estimate of the treatment effect with entropy balancing is 1.38 percentage points ($p = 0.005$) compared to our main result of 1.19 percentage points ($p = 0.012$). Moreover, we employ a Probit model to address the concern that LPM might not fit binary outcome data well. The marginal effect of this estimation is 1.28 percentage points ($p = 0.003$) and thus close to the main estimate. Third, to rule out anticipation effects, we exclude up to seven days before a climate protest (Supplementary Fig. 3a). We expect the highest, if any, anticipation just before a climate protest, as some media may report on planned protests. Such

anticipation would lead to an underestimation of the effect sizes. Estimates range from 1.23 ($p = 0.011$) to 1.64 ($p = 0.013$) percentage points. Consequently, our main estimate is a conservative finding. Fourth, we iteratively exclude climate protests to test whether the results are driven by individual protests (Supplementary Fig. 4). The estimates stay statistically significant at the five percent level, ranging from 1.02 ($p = 0.030$) to 1.63 ($p = 0.001$) percentage points. Lastly, we exclude events when two protests occurred within three days, which we count as a single protest in our main specification (Supplementary Table 6). The results do not change qualitatively (coef. = 1.17 percentage points (pp), $p = 0.016$).

Our primary outcome variable measures the concern about the consequences of climate change and is thus climate change specific. However, the SOEP also elicits concerns about environmental protection, a broader concept encompassing various environmental issues, including climate change and biodiversity loss[1,28]. We re-run our analysis using environmental protection concerns as an alternative outcome. We find a statistically significant albeit slightly smaller effect of climate protests on concerns about environmental protection (Supplementary Table 9). In our preferred specification (Column (6)), climate protests significantly increase the probability that a respondent is concerned about environmental protection on average by 0.9 percentage points ($p = 0.044$) in the 14 days after a protest. Given the protest movement's dominant focus on climate change embedded in a broader environmental discourse, this finding aligns with our expectations. It also highlights the strong but not perfect correlation between attitudes towards climate change and the environment[29]. These findings provide further evidence that concerns about climate change and the environment increase after climate protests. Further, we disaggregate the analysis and estimate the effects of each protest. Supplementary Fig. 6 displays the effect of individual climate protests on climate change concerns. While most protests yield positive coefficients, the statistical significance is sometimes reduced, not reaching the five percent significance level. This is likely due to the reduced number of observations at the protest level. Given this limitation

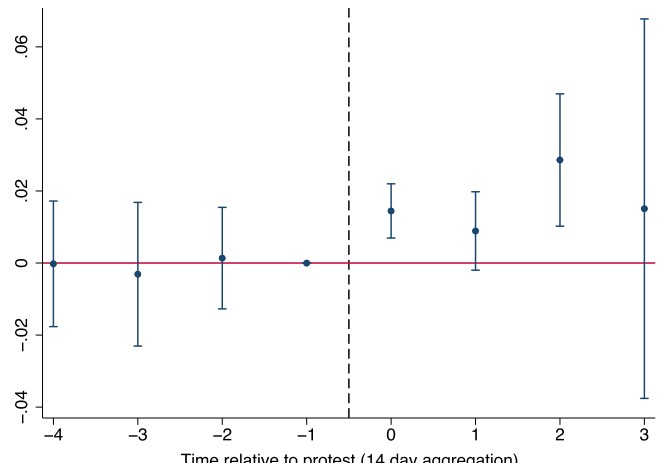

**Fig. 2 | Treatment effects over time.** Visualization of the treatment effect over time. The visualization corresponds to Column (6) of Supplementary Table 3. The number of observations for each time interval is the following: −4 (N = 3,119), −3 (N = 9,656), −2 (N = 12,692), −1 (N = 12,618), 0 (N = 11,923), 1 (N = 10,433), 2 (N = 5,993), 3 (N = 481). The confidence interval increases in the 43-56 day interval since the number of observations in later periods – that do not collide with the time window of other protests – decreases substantially. Source data are provided as a Source Data file.

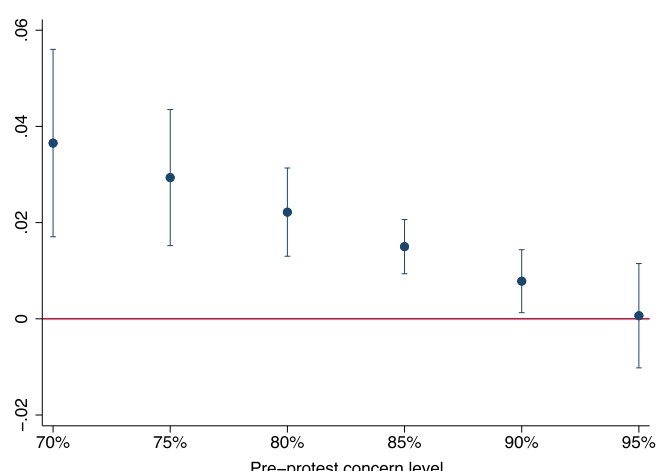

**Fig. 3 | Effects regarding pre-protest population concern level.** Plotted coefficients and 95 percent confidence intervals based on an estimated interaction term of Post with the average climate change concern in the 14 days before a protest using a linear probability model (LPM) and Specification (6) (Supplementary Table 4). Source data are provided as a Source Data file.

regarding the sample sizes, protest-specific effects should be interpreted cautiously. As a general observation, we tend to observe larger positive coefficients for the first and last protests, potentially due to comparatively lower levels of concern before these protests took place (Fig. 1) in line with the aggregate finding concerning pre-protest concern levels (see below).

Extending the time window to eight weeks, we find statistically significant coefficients at the five percent level in the 1–14 days ($p = 0.002$) and the 29–42 days ($p = 0.008$) after a protest (Fig. 2). Consequently, climate change concerns do not merely spike in the immediate aftermath of a protest. This finding is further supported by the robustness of the results to varying the time window around the treatment (Supplementary Fig. 3b). Increasing the window to up to 90 days around a protest does not substantially affect the coefficient's size (1.31pp) or statistical significance ($p = 0.003$).

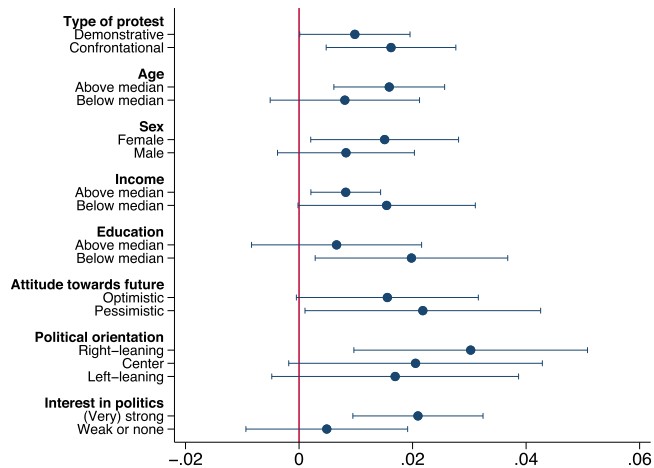

**Fig. 4 | Effects by protest movement and population subgroup.** Heterogeneous treatment effects using an LPM and Specification (6). We estimate one model per subheading and include a treatment effect (Post) per subgroup instead of the overall treatment effect. The figure plots the point estimates and the 95 percent confidence intervals. See Supplementary Table 5 for exact statistics and the number of observations of population subgroups. Source data are provided as a Source Data file.

## Concern levels prior to a protest moderate the observed effect of protests on climate concerns

The effects of protests differ depending on the average level of climate change concern before a protest. Figure 3 illustrates the estimated overall marginal effect of protests on climate change concerns in relation to the population's level of climate change concern using an interaction effect (for statistical details, please see Supplementary Table 4). The coefficient of the interaction effect is statistically significant at the five percent level ($p = 0.016$, df=24,184). It indicates that for a ten percentage point increase in concern levels before a protest, the effect size decreases by around 1.4 percentage points. It shows that the increase in concerns after protests is higher when a larger fraction of individuals are not concerned in the 14 days before a protest. This finding implies decreasing returns to protests when public climate change concerns approach high levels. The result may have two potential causes. First, as more people become concerned, it may become increasingly difficult to convince the remaining pool of dissenters[30]. A dwindling group may not be convinced by more information and awareness raised by protests, as they ideologically deny the science behind climate change[22,31]. Second, the unconcerned group might be less likely to consume national news and, therefore, less likely to be treated by climate protests. A simple positive correlation of around 0.106 ($p < 0.001$) in our data suggests that respondents not concerned about the consequences of climate change are slightly less likely to read a newspaper regularly (N = 10,689). However, even when 90 percent of the population is concerned about climate change, we find statistically significant positive impacts of climate protests on climate change concerns (coef. = 0.78pp, $p = 0.022$) at the five percent significance level.

## Climate change concerns increase following either climate strikes or civil disobedience

Critics of climate movements and their tactics argue that confrontational protests may be counterproductive to protesters' aims. Thus, we test for effect heterogeneity concerning the organizing movements and their strategies. Figure 4 divides protests by movement strategy, demonstrative protests for FFF, and confrontational acts of civil disobedience for EG and XR. We find an increase in climate change concerns following either strategy (demonstrative: coef. = 0.98pp, $p = 0.048$, df=24,184; confrontational: coef. = 1.62pp, $p = 0.008$, df =

24,184) and that the coefficients are not statistically significantly different from each other (the 95 percent confidence intervals contain the point estimates of the other protest type[32]).

We find no statistically significant evidence that climate protests negatively impact climate change concerns in different subpopulations (see Fig. 4 for the plotted coefficients). We test effects depending on age (above median: coef. = 1.59pp, $p = 0.003$; below median: coef. = 0.81pp, $p = 0.212$), sex (female: coef. = 1.51pp, $p = 0.026$; male: coef. = 0.83pp, $p = 0.165$), income (above median: coef. = 0.82pp, $p = 0.012$; below median: coef. = 1.54pp, $p = 0.052$), education (above median: coef. = 0.66pp, $p = 0.364$; below median: coef. = 1.98pp, $p = 0.025$), attitudes towards the future (optimistic: coef. = 1.56pp, $p = 0.056$; pessimistic: coef. = 2.18pp, $p = 0.041$), political orientation (right-leaning: coef. = 3.02pp, $p = 0.007$; center: coef. = 2.05pp, $p = 0.069$; left-leaning: coef. = 1.69pp, $p = 0.118$), and interest in politics ((very) strong: coef. = 2.09pp, $p = 0.001$; weak or none: coef. = 0.49pp, $p = 0.478$). A rich body of literature demonstrates that individual-level factors, ranging from socioeconomic characteristics to values and worldviews, are associated with people's beliefs and concerns about climate change[33,34]. Moreover, while Germany is less polarized than the US[35], political orientation may result in different reactions to climate protests. However, we find no statistically significant differences between effects on any population groups (age: $p = 0.231$, sex: $p = 0.425$, income: $p = 0.334$, education: $p = 0.309$, attitude towards future: $p = 0.697$, political orientation: $p = 0.572$ and $p = 0.332$, interest in politics: $p = 0.101$). Supplementary Table 5 provides detailed statistics.

Furthermore, the literature documents that civil disobedience, compared to demonstrative protests, might be particularly effective for groups that are resistant to the climate movements cause[36–40], including right-leaning groups[33,34]. We test this hypothesis by analyzing whether different protest types differ in their heterogeneous effects by subpopulation and pre-protest concern level. Supplementary Fig. 7a displays the results for pre-protest concern level by protest type, while Supplementary Fig. 7b shows whether the heterogeneous effects by political orientation differ for the two kinds of protest movements. The results should be interpreted cautiously since the sample size of certain groups is relatively small when estimating heterogeneous effects by subsets of protests. We find no conclusive evidence. Future studies with larger sample sizes may shed more conclusive light on whether distinct types of climate protests are differently effective in certain parts of the population.

## Discussion

The rise of climate change concerns in the past few years[1] has coincided with new climate movements capturing the world's attention[41]. Proponents stress protests as crucial drivers of public attention to climate change, while critics claim they are counterproductive to protesters' aims.

We find that concerns about climate change increase following protests, implying that these protests do not merely preach to the converted. Our results suggest that the analyzed protests have been an effective means to remind society of the consequences of climate change time and again. Our results further suggest that climate protests in Germany have been particularly effective when the population was not yet broadly sensitized to the consequences of climate change. Lastly, we do not find statistically significant evidence that salient climate protests negatively affect climate change concerns, irrespective of analyzing different protest tactics or subpopulations. In the context of the analyzed protests, our results thus do not provide evidence for worries of backfiring of confrontational protests related to concerns about the consequences of climate change. Our results relate to recent findings in the psychological and social movement literature observing that certain types of civil disobedience, also referred to as non-normative non-violent protests are effective in reaching the protesters' cause. For instance, studies on the US civil rights movement have shown that sit-ins have successfully influenced groups opposing the protesters' goals[36,37]. This effectiveness may be attributed to the strategic disruption caused by these protests, demonstrating society's dependence on the protesters to the resistant groups[38]. The disruptive nature of the actions in these cases may be effective by still clearly conveying the constructive objectives of the movement, a concept known as constructive disruption[39,40].

While investigating the effects of recent forms of climate protest tactics (e.g., protestors gluing themselves to streets or throwing liquids at paintings) on, for instance, public support for mitigation policies or support for the protest movement[23] would be highly relevant, we cannot speak to those outcomes with our data. Furthermore, future studies should investigate the effect of climate protests on other climate-related outcomes.

Several factors may influence the generalizability of our findings to other countries and contexts. First, as media coverage mediates the effect of climate protests, how the media frames these protests and climate change can influence their effectiveness in different countries. Notably, recent confrontational protests have led to anti-climate activism framing in various media outlets[42], aligning with Wasows's (2020)[19] findings on media's adoption of issue frames based on protest tactics. Consequently, generalizing our results to countries with more polarized media, like the US[43,44], requires caution. Second, the growing polarization around climate change[45] may further affect generalizability. A climate movement could trigger backfire effects among highly polarized population groups[22,46] if portrayed as partisan. Lastly, the characteristics and tactics of protest movements, as well as sociopolitical characteristics in different countries may influence the transferability of the findings to other contexts[10,21].

Altogether, our findings suggest that climate movements can play a role in raising public attention to climate change by organizing demonstrative or confrontational protests covered broadly in the media.

## Methods

### Climate Protest Data

We construct a database containing salient climate protests organized by various climate movements in Germany. We first identify the relevant groups organizing climate protests, focusing on the recent cycle of climate activism[11]. The groups attracting the most participants and news coverage are the youth-led Fridays for Future (FFF) and Ende Gelände (EG) (German saying for "here and no further") movements, as well as to a lesser extent the group Extinction Rebellion (XR)[47]. While FFF mobilizes global climate strikes[12], EG organizes mass actions of civil disobedience[13], such as blocking coal infrastructure[48]. XR plays a minor role in Germany, mainly focusing on acts of non-violent civil disobedience[14], such as blocking roads.

We assume that media reports are the main transmission channel for how climate protests affect the population. Sisco et al. [18] and Wasow (2020)[19] find that media attention mediates the effect of protests. Consequently, we only select climate protests that are salient to the general public through this channel. This selection is based on whether a specific climate protest is reported on in the evening news of the two leading public service broadcasters, ARD and ZDF. The Tagesschau and heute are the most trusted[49] and viewed news formats in Germany, respectively reaching 11.8 m and 4.6 m viewers in 2020[50] and are precursors for news topics in other media in Germany (see Supplementary Fig. 1). We include global climate strikes and confrontational protests in our protest database if they appeared in the reporting of either the ARD or ZDF news formats or both. Table 1 displays the climate protests identified with this method.

## German Socio-Economic Panel

The main empirical analysis relies on the SOEP, a representative longitudinal household survey of about 15,000 households conducted yearly since 1984[51]. The survey covers various topics and socioeconomic characteristics, including indicators of attitudes and concerns. The data contain information from all household members aged 12 years and over, which includes information on approximately 35,000 individuals. Our outcome variable asks, "How concerned are you about the consequences of climate change?". The categorical variable has been part of the questionnaire since 2009 and has three possible answers, ranging from no concern over a few to large concerns. Based on this variable, we construct a dummy for "concern," which equals one if a person is concerned to any extent and zero if they are not. Thus, we can measure the extensive margin of climate change concerns. Figure 1 shows the weekly averages of this variable from 2016 to 2020. We also show that our results remain the same when all three levels of the outcome variable are used (Supplementary Table 8).

## Method

The quasi-random occurrence of protests relative to survey dates makes it possible to identify the causal effect of climate protests on individuals' climate change concerns in a before-and-after research design. As the SOEP surveys respondents over the entire year, the timing may coincide with climate protests. Whether respondents are interviewed immediately before or after a protest is plausibly random. Thus, comparing responses shortly before the protest with those shortly after the protest can, under certain assumptions, identify the causal effect. This approach is also called Unexpected Event during Survey Design[25] and has been applied in a variety of settings[52,53]. In our main specification, we compare a window of 14 days before ($N = 12,633$) and after ($N = 11,933$) a protest. We present our findings using multiple alternative time windows to ensure this choice does not affect the results. Supplementary Fig. 2 demonstrates how this time window looks around an exemplary protest and identifies the treatment and control group for a protest. If the post- and pre-treatment periods of two protests overlap, we end the respective periods in the middle.

Muñoz et al. [25] argue that identification relies on two main assumptions: temporal ignorability and excludability. First, temporal ignorability implies that the timing of the survey is independent of the timing of climate protests. Thus, we must assume that comparable population groups are interviewed before and after a protest. This assumption is unlikely violated since the timing of the interview is determined by a long-standing panel survey in which the implementation logistics are decided well in advance. To assess the validity of this intuition, we conduct balance tests on respondent characteristics (age, gender, education, etc.) by regressing them on the treatment indicator and a complete set of protest dummies. The dummies ensure we compare whether the characteristics differ before and after a particular protest and not across all protests. Supplementary Table 1 presents the results of these tests, which suggest that the assumption is plausible, given no statistically significant differences in observed respondent characteristics before and after climate protests (age: $p = 0.764$, female: $p = 0.130$, household size: $p = 0.140$, education: $p = 0.895$, employment status: $p = 0.731$, teenagers in household: $p = 0.285$, household labor income: $p = 0.409$, interest in politics: $p = 0.703$, political orientation: $p = 0.172$). Excludability implies that the survey interview's timing does not impact the outcome except through the analyzed event. In our case, the timing of the SOEP interview should only affect climate change concerns through the respondents' exposure to the protest. This identifying assumption may be violated by simultaneous events or time trends in the outcome variable[25]. Although we cannot conclusively test this assumption, we provide evidence that it holds by conducting placebo tests. We test the validity by measuring i) the effect of climate protests on other concerns that

are part of the standard SOEP[54] and ii) the effect of hypothetical protests on climate change concerns (Supplementary Table 2). We do not find systematically statistically significant effects in the placebo exercise on other concerns. In particular, we test the effects on respondents' concerns about the general economic development ($p = 0.217$), own economic situation ($p = 0.071$), own pension ($p = 0.659$), own health ($p = 0.054$), peacekeeping ($p = 0.808$), job security ($p = 0.910$), immigration ($p = 0.674$), and xenophobia ($p = 0.556$). We only observe statistically significant effects on crime-related concerns (coef. = 1.09 pp, $p = 0.042$). Some statistically significant coefficients are expected when not controlling for potentially confounding events relevant to each alternative outcome and testing a broad range of hypotheses. Interestingly, climate protests may have a statistically significant impact on concerns about crime, as respondents might become more sensitive to the issues of protesters breaching the law once they are exposed to information about these protests. Importantly, we do not detect statistically significant effects of the hypothetical protests ($p = 0.484$), indicating that our method does not simply pick up time trends in climate change concerns.

We estimate the effect of protests on climate change concerns (Concern) with the following model:

$$\text{Concern}_{i,s,p,d,t} = \alpha + \beta \text{Post}_{i,p,d,t} + \gamma \mathbf{X}_{i,t} + \delta \mathbf{I}_{i,t} + \epsilon \mathbf{C}_{s,d} + \zeta_p + \eta_t + \theta \mathbf{D}_d + \varepsilon_{i,s,p,d,t}, \quad (1)$$

where $i$ denotes the individual living in state $s$, $p$ the respective protest, $d$ the date of the SOEP interview and $t$ the year. Post represents the treatment effect of protests. The dummy equals one if the individual is interviewed after the protest and zero otherwise. $\mathbf{X}_{i,t}$ is a vector of several (socioeconomic) individual-level characteristics in year $t$ that have been shown to be associated with beliefs and concerns about climate change[33,34]. We include the respondent's age, self-reported sex (dummy), number of years in education, employment status (dummy), the 2-digit industry code of the respondent's work (categorical), household size, the number of children aged 14 to 18 in the household, household labor income and the respondent's interest in politics as well as their political orientation. Political orientation is elicited on a Likert scale ranging from 0 (far left) to 10 (far right). We include a categorical variable indicating "left-leaning" (values 0–4), "right-leaning" (values 6-10), and "center" (value 5 which is the largest category). Similarly, interest in politics is elicited on a 1-4 scale, where we include "(very) strong" (values 1-2) and "weak or none" (3-4). Both variables are pre-treatment values. $\mathbf{I}_{i,t}$ is a vector of interviewer characteristics. It controls for education, sex, and age, and the variables are prepared equivalent to the respondent characteristics. To avoid the loss of observations due to missing values in these variables, we include dummies indicating missing information in a variable in $\mathbf{X}_{i,t}$ and $\mathbf{I}_{i,t}$.

We further control for external factors possibly correlated with the treatment indicator and climate change concerns in $\mathbf{C}_{s,d}$. It includes variables related to weather anomalies and relevant political events (federal elections and UN COPs). Weather data is obtained from Germany's National Meteorological Service (DWD). We operationalize temperature anomalies by taking the absolute deviation of the mean precipitation (temperature) in the month of the interview in state $s$ from the historical mean precipitation (temperature) in that state and month, and this absolute deviation of the mean precipitation (temperature) squared. Historical averages for each state and month are calculated between 1950 and 2000. The dummy for political events is equal to one in the month (week) before and after federal elections (UN COPs). $\zeta_p$ are protest fixed effects, meaning we estimate the effect of Post by comparing individuals around a particular protest. $\eta_t$ are year fixed effects and $\mathbf{D}_d$ is a vector of dummies for the day of the week and month of the year the interview took place. It controls for potential systematic differences in responses across weekdays and months since

the timing of the protest is likely correlated with certain weekdays or months (especially with FFF events). In further specifications, we also control for unobserved differences across states where the respondents live by including state fixed effects or differences across states for each protest by including state-by-protest fixed effects. In our main results, we rely on robust standard errors clustered at the level of the protests. The degrees of freedom (df) equal N-k-1[55], where k is the number of variables and equals 380 in our preferred specification resulting in df = 24,185.

### Robustness checks

To test the robustness of our results, we first show that our main results in Table 2 are independent of the chosen estimation method. In line two of Table 2, we adjust any remaining imbalances between respondents before and after protests using entropy balancing of means[27]. This method is frequently applied conjointly with the methodology used in this study[25,56]. Entropy balancing is a re-weighting scheme that calibrates unit weights to improve the balance of covariate means between the treatment and control groups further. We use all except the pre-treatment covariates to create the weights since the relatively large number of missing values would reduce the sample size (see Supplementary Table 1). In line three of Table 2, we use alternatively a Probit model instead of the Linear Probability Model to address the common concern that linear probability models might not fit binary outcome data well and predict unreasonable values outside of the zero to one range. Given that the coefficients in these models are complex to interpret, we transform the estimates into marginal effects on the probability of being concerned, making them comparable to our main estimation. Next, we adapt the time window around protests. Moreover, we exclude up to seven days before a climate protest to rule out anticipation effects (Supplementary Fig. 3a). Furthermore, we iteratively exclude climate protests to confirm that single protests do not drive the results (Supplementary Fig. 4). Lastly, we exclude events when two protests happened in three days, which we count as one in our main specification (Supplementary Table 6).

Our main results report aggregate effects across all selected climate protests. To further check the validity of our aggregate results, we analyze the individual effects of each protest. First, we split the sample into sub-samples for each protest and investigate whether the identifying assumptions of our method are still likely to hold around each protest to estimate credible effects. Supplementary Fig. 5 displays the covariate balance for each protest to test the temporal ignorability assumption for protest-specific estimations. The covariates are relatively balanced for certain protests (Supplementary Fig. 5a). However, given the reduced number of observations, the covariates of respondents around some protests are not entirely balanced. Therefore, going beyond controlling for respondent characteristics in the regressions, we adjust for remaining covariate imbalances by applying entropy balancing[27]. Supplementary Fig. 5b displays the well-balanced covariates with entropy balancing weights. Consequently, we include these weights in the protest-level estimations. We further test the excludability assumption at the protest level. Supplementary Table 7 presents the placebo tests that estimate each protest's effect on other concerns. The outcomes of these tests show that, for each protest, our empirical approach is not systematically driven by time trends in the outcome variable. These results suggest that the excludability assumption may be reasonably valid for protest-specific estimations.

The study was granted ethics approval by Hertie School's Research Ethics Officer under the application ID 20230220-27. To perform the empirical analysis, we have used Stata MP 16 64-bit (packages: reghdfe version 5.7.3, ebalance version 1.5.4, estout version 3.17, coefplot version 1.8.5, mlogit version 11.4.2, and gologit2 version 3.2.5) and R 4.3.1 (packages:.ggplot2 version 3.4.3, lubridate version 1.9.2, readtext version 0.90, dplyr version 1.1.2, gridExtra version 2.3,

haven version 2.5.3, and patchwork version 1.1.3). For further details please refer to the replication package of this study[57].

### Reporting summary

Further information on research design is available in the Nature Portfolio Reporting Summary linked to this article.

## Data availability

Source data are provided with this paper. Access to the individual-level data (SOEP-Core v37eu, https://doi.org/10.5684/soep.core.v37eu) can be requested by signing an agreement with DIW Berlin. Newspaper articles were retrieved from Dow Jones Factiva (dowjones.com/professional/factiva) and weather covariates from the Deutsche Wetterdienst (opendata.dwd.de/climate_environment/CDC). The data can be retrieved from https://doi.org/10.5281/zenodo.10451264. Source data are provided with this paper.

## Code availability

All scripts used for pre-processing and analysis are publicly available and can be retrieved from https://doi.org/10.5281/zenodo.10451264.

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

## Acknowledgements

We thank seminar participants at RWI, Hertie School, EAERE 2023, and EPG 2023, as well as Christian Flachsland, Robin Jessen, Nicolas Koch, Joelle Noailly, Steven Poelhekke, Laura Schmitz, Julius Stoll, and Malte Toetzke for their valuable feedback on earlier versions of this paper and Richard Frohn, Marlin Riede, and Moritz Odersky for their excellent research assistance. J.B. and H.G. gratefully acknowledge the financial support of the German Federal Ministry of Education and Research (BMBF) under grant 03SFK5C0 (Kopernikus Project Ariadne).

## Author contributions

J.B. and H.G. equally developed the research idea, designed the methodology, conducted the formal analysis, interpreted the results, visualized the findings, wrote the draft, and reviewed the manuscript.

## Funding

## Competing interests

The authors declare no competing interests.
