## [Peer Review File · Nature Communications]

Climate protests increase concerns about climate change:
Evidence from climate strikes and civil disobedience in
GermanyReviewers' Comments:

Reviewer #1:

Remarks to the Author:

The paper addresses an important and much debated topic using high-quality panel data from Germany. The paper is well structured and well written, and it comprises a data analysis at a very high standard.

The authors analyse their data using fixed effects for many controls, and they refer to the strong assumptions of natural experiments and provide several analyses to support these assumptions. While all this is convincing, it would be good to see a more fine-grained analysis for each of the protests considered in the analysis (including a balance test, placebo test, etc.). This might shed more light on the importance of particular protests and protests forms, which are currently hidden in the aggregate analysis. As there is no active manipulation in natural experiments and hence experimental control is very low, it would be more convincing to show the protest effects for each treatment (i.e. protest) first and then aggregate.

The placebo test seems to be selective, and the authors might want to use all items from the corresponding item battery in the SOEP. It would also be interesting to see whether the protest effects are specific for climate change as compared to environmental protection (as literature argues that climate change concern is not the same as environmental concern).

Reviewer #2:

Remarks to the Author:

I would like to thank you for the opportunity of reviewing this paper. I believe this paper provides robust results on the ongoing academic, political, and lay debate. As I am not familiar with the analytical strategy well, my review rather focused on how authors introduce the topic, how they benefitted from existing literature and previous studies, and how they summarize and discuss their findings.

- As civil obedience is a contested term in political psychology and social movement studies, a brief description of civil obedience as well as clarification for which particular protest tactics are defined as civil obedience is needed.
- On p.5, the authors discussed potential reasons for interaction effects related to average climate concerns before the protests. Even though I agree with these two potential causes, they look arbitrary; these causes should be elaborated/justified more with existing literature or relevant studies/statistics.
- Authors use the term "disruptive protests" multiple times. As what this term referring is vague and it has a bad connotation, I would suggest reconsidering using this term. If authors want to highlight the different tactics (including more radical and more pacifist ones), they can use existing concepts in social psychology and social movements literature such as normative vs. non-normative, violent vs. non-violent, confrontational vs. non-confrontational, radical vs. conventional, etc.
- Discussion can be extended. First, as the largest difference is between those interested in politics and those who are not, this subgroup heterogeneity can be discussed further related to important psycho-political concepts like political literacy and political- or self-efficacy. Second, the generalizability of the results for other nations/contexts/cultures should be discussed. In which aspects do you think these findings reflect (or not) the social/political reality of other countries? Do you think these results can be replicated in most of the countries? Why or why not? Last, a brief paragraph for (potential) limitations of this study and roadmaps for future studies can be an important addition.

Reviewer #3:

Remarks to the Author:

This paper presents research examining civilly disobedient climate protests and strikes in Germany in terms of their effectiveness in raising awareness about climate change. As a researcher focused on the psychology of protest effectiveness and civil disobedience in particular, I read this paper with great interest as there is relatively little empirical work on the effects of civil disobedience particularly in terms of climate activism, despite the growing use of this strategy. As a result, I think the findings of this paper have important theoretical and applied implications, and I would very much like to see them published. In particular I was impressed by the authors work to provide support for the causal assumptions of their model, and their efforts in engaging in different model specifications and use of different time windows as robustness checks. All that being said I did have a number of concerns I would like to see the authors address in a revision.

First, I think the observed heterogeneity of effects warrants further discussion. It seems based on the findings that effects were largest among populations that were not already concerned about climate change and among more right leaning populations (although it was not clear to me from reading the results/figures if the interaction with political ideology was significant – could the authors clarify this). I think this is important because there is emerging research (see Shuman et al., 2021; Biggs & Andrews; 2015) that the type of action studied here (sometime termed nonnormative nonviolent action) in the psychological literature is particularly effective for those more resistant to the goals of the protests. I would like to see the authors engage with this literature and the theoretical literature on why disruptive action is effective (e.g. Piven, 2008) to better contextualize their findings.

Related to this point, I was curious if the effect of type of protest depended on the target audience. According to figure 2a it seems that civil disobedience had a slightly larger effect than peaceful protests (although I am guessing this difference is not significant). I wonder however, if the authors examined an interaction between protest type (peaceful vs. civil disobedience) and pre-protest concern level or political ideology. The literature I describe above would lead one to expect that civil disobedience might be particularly effective for those with low levels of pre-protest concern and right leaning populations.

My second major point concerns generalizability. I do not think that it is an issue that the paper reports data only from Germany. Studies frequently report data only from one country, e.g. the US, and are deemed suitable for publication in top scientific outlets, so I do not think this should prevent publication. I would however like to see a more thorough discussion of how and when the authors think these results would or would not generalize. First, I think further discussion is particularly warranted in terms of their proposed mechanism. The authors suggest that media coverage drives the observed effects, but I think they need to engage more deeply with recent research (see Wasow 2020) on how media frames can determine the effects of protest. This particularly relevant for generalizability as there may be differences between how German media and media in countries with more polarized media environments frame climate protests. Such differences should be discussed as limitations on generalizability. The authors also make the claim that “these findings should encourage climate movements in countries with lower concern levels”. I agree that the moderation by pre-post concern does suggest this, but what about other factors that might be correlated with a lack of concern. Elsewhere in the paper, the authors mention that for example in the US the issue of climate change is more politicized and moralized. I would like to see a more deep engagement in how polarization around the issue of climate change might effect generalizability.

Finally, the authors note that they did not find any evidence of backfire effects. However, I think this claim bears some qualification. The authors did not find evidence of backfire on the outcome variable they studied, namely concern. However, past research would suggest that the same type of action can

have different effects on different outcome variables. For example, Shuman et al., 2021 found that civilly disobedient actions are the most effective type of action for increasing support for policy change (compared to completely peaceful normative protests and to violent protests), but Feinberg et al., 2020 found that more radical civilly disobedient protests decrease willingness to express support and participate in the movement. If the authors have other variables in their dataset that can speak to the effects on other variables this would help address this. If not, they should acknowledge that they cannot rule out backfire effects on other potentially relevant variables.

In sum, I found this paper an interesting engaging presentation of rigorous and important research. While I did have concerns, I think they can be addressed through revisions and I would like to see a revised version of this paper published.

**Authors' Response to Reviewers on
"Climate protests increase concerns about climate change"
Ms. Ref. No.: NCOMMS-23-14817**

We are indebted to the three anonymous reviewers for their valuable feedback. We have thoroughly revised our manuscript and added additional statistical analyses in response to their comments and feedback. First, we added a detailed analysis of individual protests comprising the aggregated data. Second, we added the proposed additional analyses, including a heterogeneity analysis by protest strategy and environmental concerns as an alternative outcome. Third, we have improved the manuscript by adding a thorough discussion of the concepts used and linking it back to different literatures, from social psychology to social movement studies. Moreover, we have added an extended discussion of the generalizability and limitations of this study. Finally, we have included all the specific comments raised by the reviewers. We believe that these revisions have resulted in a more insightful and accessible article.

The following points respond to each reviewer's comments in detail. For this purpose, the original reviewer comments have been italicized, while our responses are shown in regular font.

Reviewer #1

The paper addresses an important and much debated topic using high-quality panel data from Germany. The paper is well structured and well written, and it comprises a data analysis at a very high standard.

- *The authors analyse their data using fixed effects for many controls, and they refer to the strong assumptions of natural experiments and provide several analyses to support these assumptions. While all this is convincing, it would be good to see a more fine-grained analysis for each of the protests considered in the analysis (including a balance test, placebo test, etc.). This might shed more light on the importance of particular protests and protest forms, which are currently hidden in the aggregate analysis. As there is no active manipulation in natural experiments and hence experimental control is very low, it would be more convincing to show the protest effects for each treatment (i.e. protest) first and then aggregate.*

We appreciate the valuable suggestion the reviewer provided, which helped us significantly improve the manuscript. We have incorporated an additional subsection in Methods titled "Analysis of individual protests". Within this section, we discuss and present protest-specific balance tests, placebo tests, and their corresponding effects. These additions enhance the robustness and depth of our analysis, strengthening the overall findings of our study.

As suggested by Muñoz et al. (2020), when using our method on small subsamples, we adjust any protest-specific imbalances between respondents before and after protests using entropy balancing (Hainmüller, 2012). This re-weighting scheme calibrates unit weights to improve the balance between the treatment and control groups and is particularly suitable for small and imbalanced samples. To demonstrate the comparability of our subsample analyses with our aggregate results, we included an additional row in Table 2a, where entropy balancing is applied also to our aggregate sample. The findings remain qualitatively similar; however, there is a notable increase in the statistical significance.

The protest-specific results should be interpreted cautiously given the relatively small sample size of some individual protests. However, the main findings do not change substantially. The overall

consistency of positive coefficients across most protests and specifications further strengthens our main findings derived from the aggregate analysis.

The new section also discusses potential heterogeneity across protests in more detail, mainly highlighting some patterns underlying the aggregate results. However, while the individual protest effect sizes may be cautiously interpreted and rationalized, we refrained from adding the interpretation of specific individual protests to the paper, as we feel there is too much uncertainty given the reduced sample size and statistical power. Here, we scrutinize the smallest and one of the largest effect sizes of individual protests. The large effect of Protest 3 may be due to government coalition talks during that time. In these discussions, climate issues played an important role that may have influenced the public's susceptibility to climate change news. The negative effect of Protest 4 may be explained by the *Hambach Forest* protests being a long-term blockade, where an activist died shortly before the demonstrations. News coverage of the protest movement before the large protest might have led to an underestimation of the treatment effect.

- *The placebo test seems to be selective, and the authors might want to use all items from the corresponding item battery in the SOEP. It would also be interesting to see whether the protest effects are specific for climate change as compared to environmental protection (as literature argues that climate change concern is not the same as environmental concern).*

We appreciate the feedback and have considered it by implementing several improvements in our study. First, we incorporated the full battery of concerns available in the SOEP (Rohrer et al., 2021) into the placebo exercise, the results of which are shown in Supplementary Tab. 2. These additions are discussed in more detail in the methods section. Furthermore, we are grateful for the second suggestion to explore the impact of climate protests on concerns about environmental protection. In response, we have re-run our analysis and included the results of this secondary outcome in Supplementary Tab. 7. We find statistically significant, albeit slightly smaller, effects on concerns about environmental protection. Notably, they are still slightly larger than the effects of dry spells and temperature anomalies on environmental concerns found by Hoffmann et al. (2022). Given the central focus of the protest movement on climate change within a broader environmental discourse, the finding align with our expectations. It also underscores the strong correlation between concerns regarding the consequences of climate change and broader environmental concerns (Hornsey et al., 2016). We have included a dedicated paragraph titled "Environmental concern as an alternative outcome" in the Methods section to elaborate on these findings. These additions allow for a more comprehensive understanding of the relationship between protests, concerns regarding climate change, and broader environmental protection.

Reviewer #2

I would like to thank you for the opportunity of reviewing this paper. I believe this paper provides robust results on the ongoing academic, political, and lay debate. As I am not familiar with the analytical strategy well, my review rather focused on how authors introduce the topic, how they benefitted from existing literature and previous studies, and how they summarize and discuss their findings.

- *As civil obedience is a contested term in political psychology and social movement studies, a brief description of civil obedience as well as clarification for which particular protest tactics are defined as civil obedience is needed.*

We appreciate this feedback and have incorporated the suggested revisions into our study. Specifically, we have included a clear explanation in the Introduction, outlining our definition of civil disobedience and specifying the protest tactics we consider within this concept. This addition aims to provide readers with a better understanding of the scope and criteria used to identify and classify civil disobedience in this research.

- *On p.5, the authors discussed potential reasons for interaction effects related to average climate concerns before the protests. Even though I agree with these two potential causes, they look arbitrary; these causes should be elaborated/justified more with existing literature or relevant studies/statistics.*

We thank the reviewer for their comment. We have weakened the text ('...may have two potential causes') and we have justified the two potential causes in the text.

For the first suggested cause, we have added supporting literature to the manuscript. As more people become concerned, it may become increasingly difficult to convince the remaining pool of dissenters (McDermott, 2021). A dwindling group may not be convinced through more information and awareness from protests, as they ideologically deny the science behind climate change (Hart & Nisbet, 2012; Ma et al., 2018).

For the second suggested cause, we calculated the correlation between climate change concerns and newspaper consumption in our data. Specifically, we generate a dummy equal to one if a respondent reads the newspaper (including e-newspapers) at least once a month and zero otherwise, and calculate the correlation with our binary outcome. A simple positive correlation of around 0.11 suggests that respondents not concerned about the consequences of climate change are slightly less likely to regularly read a newspaper (N=10,689).

- *Authors use the term "disruptive protests" multiple times. As what this term referring is vague and it has a bad connotation, I would suggest reconsidering using this term. If authors want to highlight the different tactics (including more radical and more pacifist ones), they can use existing concepts in social psychology and social movements literature such as normative vs. non-normative, violent vs. non-violent, confrontational vs. non-confrontational, radical vs. conventional, etc.*

We thank the reviewer for the valuable feedback. In response, we have extensively explored the suggested concepts within the realms of social psychology and social movement literature and adjusted the text accordingly. Considering the reviewer's comment, we have refrained from defining protests as "disruptive," as we acknowledge that this term carries a negative connotation and lacks a clear, universally accepted definition. Instead, we have adopted the proposed distinction between *confrontational* and *non-confrontational/demonstrative* protests. This differentiation aligns with the arguments Kriesi et al. (1995) and Rucht (2007) put forth. Notably, this categorization has been employed in recent publications (Gattinara et al., 2022; Loukakis & Portos, 2020) and in the PolDem protests database, classifying protests across Europe (Kriesi et al., 2020).

- *Discussion can be extended. First, as the largest difference is between those interested in politics and those who are not, this subgroup heterogeneity can be discussed further related to important psycho-political concepts like political literacy and political- or self-efficacy. Second, the generalizability of the results for other nations/context/cultures should be discussed. In which aspects do you think these findings reflect (or not) the social/political reality of other countries? Do you think these results can be replicated in most of the*

countries? Why or why not? Last, a brief paragraph for (potential) limitations of this study and roadmaps for future studies can be an important addition.

We sincerely appreciate the valuable feedback provided by the reviewer. In response to the first point, we have expanded the discussion on the relationship between political interest and psychopolitical concepts such as political- and self-efficacy within the text. Individuals' political literacy and political- or self-efficacy may influence how effective protests are in changing the respondents' views on climate change (Landmann & Rohmann, 2020; Doherty & Webler, 2016). The positive correlation of around 0.19 in our data suggests that respondents who are interested in politics are more likely to (strongly) support the notion that social engagement can influence social conditions (N=9,150).

Furthermore, we thank the reviewer for their guidance on the generalizability of our findings. Following this feedback and the comments from Reviewer #3, we have included a new paragraph in the Discussion section that extensively addresses the generalizability of our results. In this paragraph, we identify three significant factors that may limit the generalizability of our findings to other settings. We also discuss the potential implications of these limitations and offer insights into their potential impact on the broader applicability of our study. This clarification highlights the boundaries of this study and its applicability to specific contexts. In addition, we have added sentences to different sections of the paper and a paragraph in the Discussion, suggesting future research directions. These recommendations aim to guide future studies by identifying areas that require further investigation.

Reviewer #3

This paper presents research examining civilly disobedient climate protests and strikes in Germany in terms of their effectiveness in raising awareness about climate change. As a researcher focused on the psychology of protest effectiveness and civil disobedience in particular, I read this paper with great interest as there is relatively little empirical work on the effects of civil disobedience particularly in terms of climate activism, despite the growing use of this strategy. As a result, I think the findings of this paper have important theoretical and applied implications, and I would very much like to see them published. In particular I was impressed by the authors work to provide support for the causal assumptions of their model, and their efforts in engaging in different model specifications and use of different time windows as robustness checks. All that being said I did have a number of concerns I would like to see the authors address in a revision.

- *First, I think the observed heterogeneity of effects warrants further discussion. It seems based on the findings that effects were largest among populations that were not already concerned about climate change and among more right leaning populations (although it was not clear to me from reading the results/figures if the interaction with political ideology was significant – could the authors clarify this). I think this is important because there is emerging research (see Shuman et al., 2021; Biggs & Andrews; 2015) that the type of action studied here (sometime termed nonnormative nonviolent action) in the psychological literature is particularly effective for those more resistant to the goals of the protests. I would like to see the authors engage with this literature and the theoretical literature on why disruptive action is effective (e.g. Piven, 2008) to better contextualize their findings.*

We thank the reviewer for the insightful and helpful comments. Indeed, while the coefficient for political ideology is positive for all groups (left-leaning, center, right-leaning), the largest coefficient and the only one that is statistically significant at the 5 percent level is found for right-leaning

individuals. However, the coefficient is not statistically significantly different from those for other political orientations, which we clarified in the text.

Indeed, we find indicative, although not conclusive, evidence that the protests we analyze seem to be particularly effective among the groups most resistant to the protest's goals. We have added a paragraph to the Results section dedicated to discussing these findings in light of the literature strand pointed out by the reviewer. It introduces concepts such as nonnormative nonviolent action and constructive disruption and emphasizes that our findings align with those from different contexts.

- *Related to this point, I was curious if the effect of type of protest depended on the target audience. According to figure 2a it seems that civil disobedience had a slightly larger effect than peaceful protests (although I am guessing this difference is not significant). I wonder however, if the authors examined an interaction between protest type (peaceful vs. civil disobedience) and pre-protest concern level or political ideology. The literature I describe above would lead one to expect that civil disobedience might be particularly effective for those with low levels of pre-protest concern and right leaning populations.*

We thank the reviewer for the great suggestion. Indeed, while civil disobedience protests have a larger point estimate, the effects by type of protest are not statistically significant from each other. We have made this more transparent in the revised manuscript.

We implemented the suggested interaction, refer to it in the Results section, and discuss it in more detail in the method section under the new subheading “Heterogeneous effects by type of protest”. We implemented a triple interaction between the post-protest indicator, the type of protest (demonstrative vs. confrontational), and the pre-protest concern level (Supplementary Fig. 8a) or political ideology (Supplementary Fig. 8b). The results of these estimations should be interpreted with caution, since the sample size in this setting for that many interactions with individual categories is relatively small. We find results that align with the aggregate results but no evidence for significant differences between the effects of the different protest types for low levels of pre-protest concern or right-leaning populations. Future studies with more observations might shed more conclusive light on whether different types of climate protests are differently effective in certain parts of the population.

- *My second major point concerns generalizability. I do not think that it is an issue that the paper reports data only from Germany. Studies frequently report data only from one country, e.g. the US, and are deemed suitable for publication in top scientific outlets, so I do not think this should prevent publication. I would however like to see a more thorough discussion of how and when the authors think these results would or would not generalize. First, I think further discussion is particularly warranted in terms of their proposed mechanism. The authors suggest that media coverage drives the observed effects, but I think they need to engage more deeply with recent research (see Wasow 2020) on how media frames can determine the effects of protest. This particularly relevant for generalizability as there may be differences between how German media and media in countries with more polarized media environments frame climate protests. Such differences should be discussed as limitations on generalizability. The authors also make the claim that “these findings should encourage climate movements in countries with lower concern levels”. I agree that the moderation by pre-post concern does suggest this, but what about other factors that might be correlated with a lack of concern. Elsewhere in the paper, the authors mention that for example in the US the issue of climate change is more politicized and moralized. I would like*

to see a more deep engagement in how polarization around the issue of climate change might effect generalizability.

We are deeply grateful to the reviewer for the comments and pointers to the facets that may influence the generalizability of the study's findings. In response to these comments and the comments from Reviewer #2, we have made significant improvements to our manuscript. Specifically, we have included a new paragraph in the discussion section that delves deeper into generalizability, addressing three key aspects: media issue frames and media polarization, general political polarization and polarization related explicitly to climate change, and the socio-political characteristics and structure of climate movements in different countries. By incorporating this additional paragraph, we aim to provide readers with a more comprehensive understanding of the factors that may affect the generalizability of our findings, as well as the limitations associated with applying our results to other countries and contexts.

- *Finally, the authors note that they did not find any evidence of backfire effects. However, I think this claim bears some qualification. The authors did not find evidence of backfire on the outcome variable they studied, namely concern. However, past research would suggest that the same type of action can have different effects on different outcome variables. For example, Shuman et al., 2021 found that civilly disobedient actions are the most effective type of action for increasing support for policy change (compared to completely peaceful normative protests and to violent protests), but Feinberg et al., 2020 found that more radical civilly disobedient protests decrease willingness to express support and participate in the movement. If the authors have other variables in their dataset that can speak to the effects on other variables this would help address this. If not, they should acknowledge that they cannot rule out backfire effects on other potentially relevant variables.*

We thank the reviewer for this comment. We have amended the text and clarified that only for climate change concerns, there is no evidence of backfire effects. Given Reviewer #1's comment, we have added concerns regarding environmental protection as a secondary outcome (Supplementary Tab. 8). While this is a very similar outcome, it further strengthens the robustness of our findings for a slightly different outcome. Unfortunately, data limitations hinder the investigation of relevant outcome variables related to public support for mitigation policies or support for the protest movement (Feinberg et al., 2020). Consequently, we have acknowledged in the manuscript that we cannot speak to these outcomes and that future research is needed.

Reviewers' Comments:

Reviewer #1:

Remarks to the Author:

The authors considered all of my points, and in my view the paper is significantly improved. I have only one substantial remark: I would like to see the effects of protests on crime related and economic concerns to be discussed in the main text and not the methods section/supplementary material. If I understand the results correctly this can be very interesting for the social movement literature, considering a careful interpretation, of course.

Reviewer #2:

Remarks to the Author:

I wish to thank to authors, as all my concerns and suggestions are carefully addressed. I read it thoroughly and I believe that the article has made significant progress both in terms of presenting the current literature and research question more clearly and in terms of more detailed and careful discussion of the findings. In conclusion, I believe this version represents a significantly stronger contribution to multiple literature. I am looking forward to reading its published version.

Reviewer #3:

Remarks to the Author:

The authors satisfactorily addressed my concerns and I am looking forward to seeing this paper published!

**Authors' Response to Reviewers on
"Climate protests increase concerns about climate change"
Ms. Ref. No.: NCOMMS-23-14817A**

We thank the three anonymous reviewers for their valuable feedback and remarks, which improved the paper significantly.

The following points respond to each reviewer's comments in detail. For this purpose, the original reviewer comments have been italicized, while our responses are shown in regular font.

Reviewer #1 (Remarks to the Author):

The authors considered all of my points, and in my view the paper is significantly improved. I have only one substantial remark: I would like to see the effects of protests on crime related and economic concerns to be discussed in the main text and not the methods section/supplementary material. If I understand the results correctly this can be very interesting for the social movement literature, considering a careful interpretation, of course.

We thank the reviewer for the valuable feedback helping us to improve the manuscript. Following the reviewer's suggestion, we have now also incorporated the placebo result for crime in the main text alongside the result for environmental concerns (line 67-72). Following the statistical reporting guidelines of Nature journals, we have refrained from adding the placebo results for own economic situation in the main text since the placebo test does not reach conventional levels of statistical significance (5%) in the full specification. We have also adjusted the text in the Methods section accordingly.

Reviewer #2 (Remarks to the Author):

I wish to thank to authors, as all my concerns and suggestions are carefully addressed. I read it thoroughly and I believe that the article has made significant progress both in terms of presenting the current literature and research question more clearly and in terms of more detailed and careful discussion of the findings. In conclusion, I believe this version represents a significantly stronger contribution to multiple literature. I am looking forward to reading its published version.

We thank the reviewer for the warm words and for providing useful feedback on how to sharpen the arguments, embed the article in the current literature and discuss the findings.

Reviewer #3 (Remarks to the Author):

The authors satisfactorily addressed my concerns and I am looking forward to seeing this paper published!

We would like to express our gratitude to the reviewer for helping us improve the paper!